# Correlation of Mortality Burdens of Cerebrovascular Disease and Diabetes Mellitus with Domestic Consumption of Soya and Palm Oils

**DOI:** 10.3390/ijerph17155410

**Published:** 2020-07-28

**Authors:** Maznah Ismail, Abdulsamad Alsalahi, Huzwah Khaza’ai, Mustapha Umar Imam, Der Jiun Ooi, Mad Nasir Samsudin, Zulkifli Idrus, Muhammed Ha’iz Mohd Sokhini, Musheer A. Aljaberi

**Affiliations:** 1Laboratory of Molecular Biomedicine, Institute of Bioscience, Universiti Putra Malaysia, UPM 43400, Serdang, Selangor, Malaysia; ahmedsamad28@yahoo.com; 2Department of Pharmacology, Faculty of Pharmacy, Sana’a University, Mazbah District, Sana’a Secretariat 1247, Yemen; 3Department of Biomedicine, Faculty of Medicine and Health Sciences, Universiti Putra Malaysia, UPM 43400, Serdang, Selangor, Malaysia; huzwah@upm.edu.my; 4Centre for Advanced Medical Research and Training, Usmanu Danfodiyo University, Sokoto 840231, Nigeria; mustyimam@gmail.com; 5Department of Medical Biochemistry, Faculty of Basic Medical Sciences, Usmanu Danfodiyo University, Sokoto 840231, Nigeria; 6Department of Oral Biology & Biomedical Sciences, Faculty of Dentistry, MAHSA University, Jenjarom Selangor 42610, Malaysia; djooi@mahsa.edu.my; 7Department of Agribusiness and Bioresource Economics, Faculty of Agriculture, Universiti Putra Malaysia, UPM 43400, Serdang, Selangor, Malaysia; mns@upm.edu.my; 8Institute of Tropical Agriculture and Food Security, Universiti Putra Malaysia, UPM 43400, Serdang, Selangor, Malaysia; zulidrus@upm.edu.my; 9Ethical Classic Business, Duopharma Marketing Sdn. Bhd. Lot No 2,4,6,8 & 10, Jalan P/7, Seksyen 13, Kawasan Perusahaan, Bandar Baru Bangi 43650, Selangor, Malaysia; haizsokhini@doupharmabiotech.com; 10Community Health Department, Faculty of Medicine and Health Sciences, Universiti Putra Malaysia, UPM 43400, Serdang, Selangor, Malaysia; gabrisyria@yahoo.com

**Keywords:** cerebrovascular diseases, diabetes mellitus, palm oil, soya oil

## Abstract

Background: Cerebrovascular diseases (CBVDs) and diabetes mellitus (DM) are interrelated and cumbersome global health burdens. However, the relationship between edible oils consumption and mortality burdens of CBVDs and DM has not yet been evaluated. This review aims to explore correlations between per capita mortality burdens of CBVDs and DM, as well as food consumption of palm or soya oils in 11 randomly selected countries in 2005, 2010, and 2016. Methods: After obtaining data on food consumption of palm and soya oils and mortality burdens of CBVDs and DM, correlations between the consumption of oils and mortality burdens of diseases were explored. Results: There was a positive correlation between the consumption of soya oil with the mortality burden of CBVDs in Australia, Switzerland, and Indonesia, as well as the mortality burden of DM in the USA. The consumption of palm oil had a positive correlation with the mortality burden of DM in Jordan only. Conclusions: Food consumption of soya oil in several countries possibly contributes to the mortality burden of CBVDs or DM more than food consumption of palm oil, which could be a possible risk factor in the mortality burdens of CBVDs and DM.

## 1. Introduction

Cerebrovascular diseases (CBVDs) are diverse and multiple conditions of neurological disorders involving the blood vessels of the brain [1,2] which include ischemic strokes, intracerebral hemorrhagic strokes, aneurysms, arteriovenous malformations, heart arrests, as well as occluded and stenotic carotid-arteries that lead to cognitive disorders, namely, vascular cognitive impairment and vascular dementia [2,3].

In 2016, the World Health Organization reported that CBVDs occupy the second-largest single cause of death in the world and Europe [4], the fifth most common cause of death in the USA [5], and the second leading cause of death worldwide [6], while CBVD with brain hypoperfusion is a strong risk factor for stroke [7]. In 2010, 16.9 million people suffered from a stroke, which was the first-ever stroke epidemic worldwide [8]. 

Several risk factors are associated with CVBDs, including diabetes mellitus (DM) [5], while Type 2 diabetes mellitus (T2DM) itself is one of the most cumbersome global chronic diseases, which is also associated with morbidity and mortality [9]. Approximately 8.8% of the global population were diabetic in 2017, and this could reach nearly 10% by 2045 [10]. In addition, DM is associated with micro- and macrovascular disorders, of which CBVD is one of those microvascular disorders [11]. Moreover, the chronic vascular complications are the major causes of death of T2DM [12]. Accordingly, such association suggests the evaluation of the mortality burdens of DM side by side with that of CBVDs.

There is a health gap in the mortality data of CBVDs between the USA, Canada, and Latin American countries, as well as between western and eastern European countries, which could be due to lifestyle factors, diet, and socioeconomic influences [13]. In addition, silent CBVD leads to serious risks such as stroke and dementia [14]. Moreover, there is an increasing interest in the indices of total CBVDs, incorporating multiple lesions, as conventional markers of CBVD [15]. Accordingly, it necessitates conducting investigations to assess the causes that stand behind the mortality burdens of total CBVD [4], which constitutes a greater challenge to be assessed since the abnormalities in cerebrovasculature do not result in immediate symptoms [8].

According to the report of the WHO (2018), a significant gain in global human longevity has been observed in the last couple of decades, as evidenced by an average 5.5-year increase in life expectancy between 2000 and 2016 [16], whereas the average life expectancy exceeds 80 years old in many countries [16]. Longevity gains have come at a cost, however, with the most obvious being an increase in age-related diseases [17], including noncommunicable diseases (NCDs) such as DM, musculoskeletal disorders, cardiovascular diseases, and neurological disorders, which place a burden on individuals and healthcare systems [18]. Preventing primary hypertension, smoking, DM, and atrial fibrillation should be considered since preventing these risk factors could reduce the incidence of CBVDs [8]. However, stroke remains a major cause of death and disability, and therefore research should be continued [8].

Since the high intake of saturated fatty acid correlates with increased levels of cholesterol and hence cardiovascular disorders, this suggests that it is an important risk factor for CBVDs as well since CBVD is a sequela of cardiovascular disease [19]. There are several vegetable oils that are well-known for their fat content, including palm and soya oils, of which consumption should be evaluated to assess whether their consumption could be associated with CBVDs or DM. In addition, there was an epidemiological study that indicated that the increase in the consumption of palm oil was related to the higher mortality rate of ischemic heart disease in developing countries from 1980 to 1997 [20]. However, recent reports indicated that dietary palm oil exerts a neuroprotective effect in cells and animals [21]. In addition, the World Health Organization (2003) reported that there is no convincing evidence that palm oil consumption contributes to an increased risk of developing cardiovascular diseases [19]. Furthermore, the simulation indicated that the consumption of palm oil in cooking rather than other dietary oils was associated with negative health burdens in Thailand [22]. Nevertheless, there has been a gap of knowledge in the literature about the possible relation between the food consumption of palm and soya oils with the mortality burden of CBVDs or DM. Accordingly, this study focuses on the correlation between the mortality burden of either CBVDs or DM, with the food-use consumption of either palm or soya oil in 11 randomly selected countries worldwide in 2005, 2010, and 2016.

## 2. Materials and Methods 

This study was based on using quantitative data in reports, documents, and records from databases about mortality burdens of either CBVDs and DM, as well as food-use consumption of either palm or soya oils. Hence, this study is not an experimental or field study and does not need ethical approval for obtaining the data, and personally identifiable information about individuals was not required.

The annual per capita mortality burden of either CBVDs or DM indicates the annual number of death cases per 100,000 population due to cerebrovascular disease or diabetes mellitus in 2005, 2010, and 2016, while the annual food-use consumption of either palm or soya oil indicates the amount of each oil in tons that was used to make food per capita in 2005, 2010, and 2016.

### 2.1. Data Sources

Data for the annual per capita consumption of palm and soya oils (in tons) for foods were obtained from an online, publicly available source [23], while data for annual per capita mortality burdens of CBVDs and DM, as well as the ranking of countries according to income, were obtained as an Excel spreadsheet [24].

### 2.2. Selection Criteria

The timeframe of mortality burdens and consumption of oils were selected to include 2005, 2010, and 2016 according to data availability, while the in-between years were excluded because of some missing data. The countries were selected randomly by an independent party using a simple random sampling method after encoding the countries by symbols in an excel spreadsheet. Then, the eligible countries were included in the study. The 11 selected countries were classified as high-income countries (the USA, Japan, Australia, and Switzerland), high middle-income countries (Turkey, Saudi Arabia, and Malaysia), and middle-income countries (Brazil, China, Jordan, and Indonesia) [24].

### 2.3. Calculations

#### 2.3.1. Standardization of Data

Data of per capita mortality burdens of either CBVDs or DM were included in this study as age- and risk-standardized death rate per 100,000 population [24], which is defined as a weighted average of the age-specific mortality rates per 100,000 persons, where the weights are the proportions of persons in the corresponding age groups of the WHO standard population [25], while the risk-standardized death rate is defined as the ratio of the number of “predicted” to “expected” deaths, multiplied by the patient-level raw mortality rate for the dataset [26]. The risk factors that should be cleared include the exposure to food-use consumption of palm and soy oils, particularly that diet is a potentially modifiable factor influencing CBVDs and DM. The values of food-use consumption of oils were obtained from the source [23] as the overall annual per capita food-use consumption of oils in tons.

#### 2.3.2. Percentage Changes

The trend of changes (increase or decrease) in the mortality burdens of diseases and food-use consumption of oils between 2005 to 2010 or 2010 to 2016 are expressed in percentages, which was adapted according to Investopedia [27].
(1)$$\mathrm{Percentage}\;\mathrm{changes}\;=\frac{value\;in\;2010-value\;in\;2005}{vlaue\;in\;2005}\;\times 100$$
(2)$$\mathrm{Percentage}\;\mathrm{changes}\;=\frac{value\;in\;2016-value\;in\;2010}{vlaue\;in\;2010}\;\times 100$$
(1) The formula to calculate the percentage change of mortality burden of a disease or food-use consumption of oils between 2005 and 2010.(2) The formula to calculate the percentage change of mortality burden of a disease or food-use consumption of oils between 2010 and 2016.

If the percentage change is negative, it means a decrease in the mortality burden of a disease or the food-use consumption of oils. If the percentage change is positive, it means an increase in the mortality burden of a disease or the food-use consumption of oils.

### 2.4. Statistical Analysis

Statistical analysis was performed using the SPSS software package (version 23, IBM). The correlation between the set of data of the mortality burdens of each disease and food-use consumption of each oil in 2005, 2010, and 2016 was performed using nonparametric Spearman’s correlation, expressed as correlation coefficient “r”, and *p* ≤ 0.05 to indicate a significant positive or negative correlation.

## 3. Results

### 3.1. Trends of Change in Mortality Burdens of CBVDs and DM Against Food-Use Consumption of Oils with Time

#### 3.1.1. The Trend of Changes in the Mortality Burden and Food-Use Consumption of Oils in the 11 Countries by Time

There was a downward trend for the mortality burdens of CBVDs and DM between 2005, 2010, and 2016. On the other hand, there was a downward trend for consumption of palm oil from 2005 to 2010 and an upward trend from 2010 to 2016, while there was an upward trend in consumption of soya oil between 2005, 2010, and 2016.

#### 3.1.2. The Trend of Changes in the Mortality Burden and Food-Use Consumption of Oils in the 11 Countries by Country Strata

##### In High-Income Countries

Mortality burdens of CBVDs in Japan, Australia, and Switzerland decreased between 2005 and 2016, while the mortality burden of CBVDs in the USA remained relatively unstable (decreased between 2005 and 2010, followed by an increase between 2010 and 2016). On the other hand, the mortality burdens of DM in the USA, Japan, Australia, and Switzerland decreased between 2005 and 2016 (Figure 1a).

Consumption of palm oil in the USA, Japan, and Australia increased between 2005 and 2016, while the consumption of palm oil in Switzerland decreased between 2005 and 2010 but increased from 2010 to 2016. The consumption of soya oil in the USA, Australia, and Switzerland decreased from 2005 to 2016, while, in Japan, the consumption of soya oil decreased from 2005 to 2010 and then appeared to stabilize (increased between 2010 and 2016; Figure 1b).

##### In High Middle-Income Countries

The mortality burdens of CBVDs and DM in Saudi Arabia and Malaysia decreased between 2005 and 2016. In Turkey, the mortality burden of CBVDs increased from 2005 to 2010 but decreased from 2010 to 2016. Unlike CBVDs, the mortality burden of DM in Turkey decreased from 2005 to 2010 but increased from 2010 to 2016 (Figure 2a).

The consumption of palm oil in Turkey and Saudi Arabia decreased from 2005 to 2010, but increased from 2010 to 2016, while the consumption of palm oil in Malaysia increased between 2005 and 2016. Conversely, the consumption of soya oil in Saudi Arabia and Malaysia increased between 2010 and 2016. In Turkey, the consumption of soya oil decreased between 2005 to 2016 (Figure 2b).

##### In Middle-Income Countries

The mortality burdens of CBVDs and DM in Brazil, China, and Jordan decreased from 2005 to 2016, in contrast to Indonesia, where these burdens of CBVDs and DM increased during that period (Figure 3a).

The consumption of palm oil in Brazil and Indonesia increased between 2005 and 2016, in contrast to Jordan, where this consumption decreased during that period. However, the consumption of palm oil in China increased from 2005 to 2010 but decreased from 2010 to 2016. Unlike palm oil, the consumption of soya oil in Brazil, China, and Indonesia increased between 2005 and 2016. In Jordan, the domestic consumption of soya oil decreased from 2005 to 2010 but increased from 2010 to 2016 (Figure 3b).

### 3.2. Correlation of Mortality Burdens of Cerebrovascular Diseases or Consumption of Oils with Time

#### 3.2.1. In High-Income Countries

In the USA and Australia, the mortality burdens of CBVDs and DM showed nonsignificant correlations with time. In Japan, the mortality burdens of CBVDs and DM showed significant negative correlations with time (*p* ≤ 0.028 and *p* ≤ 0.009, respectively). In Switzerland, the mortality burden of CBVDs showed a significant (*p* ≤ 0.023) negative correlation with time, while the mortality burden of DM showed a nonsignificant correlation with time (original data are shown in Table 1).

In the USA and Australia, the consumption of palm and soya oils showed nonsignificant correlations with time. In Japan, only the consumption of palm oil showed a significant (*p* ≤ 0.014) positive correlation with time, while the consumption of soya oil showed a nonsignificant correlation with time. In Switzerland, only the consumption of soya oil showed a significant negative correlation with time (*p* ≤ 0.001), while the consumption of palm oil showed a nonsignificant correlation with time (original data are shown in Table 1).

#### 3.2.2. In High Middle-Income Countries

In Turkey and Malaysia, the mortality burdens of CBVDs and DM showed nonsignificant correlations with time. In Saudi Arabia, the mortality burden of DM showed significant (*p* ≤ 0.014) negative correlations with time, while the mortality burden of CBVDs showed a nonsignificant correlation with time (original data are showed in Table 2).

In Turkey and Malaysia, the consumption of palm and soya oils showed a nonsignificant correlation with time (Table 2). In Saudi Arabia, the consumption of soya oil showed a significant (*p* ≤ 0.017) positive correlation with time, while the consumption of palm oil showed a nonsignificant correlation with time (original data are shown in Table 2).

#### 3.2.3. In Middle-Income Countries 

In Brazil, the mortality burden of CBVDs showed a significant negative correlation with time (*p* ≤ 0.008), while the mortality burden of DM showed a nonsignificant correlation with time. In China, Jordan, and Indonesia, the mortality burdens of CBVDs and DM showed a nonsignificant correlation with time (original data are shown in Table 3).

In Brazil and China, the consumption of soya oil showed a significant positive correlation with time (*p* ≤ 0.038 and *p* ≤ 0.049, respectively), while the consumption of palm oil showed a nonsignificant correlation with time. In Jordan, neither the consumption of palm nor soya oil showed a significant correlation with time. In Indonesia, the consumption of palm oil showed a significant positive correlation with time (*p* ≤ 0.001), while the consumption of soya oil showed a nonsignificant correlation with time (original data are shown in Table 3). 

### 3.3. Correlation between Consumption of Oils and Mortality Burdens of Diseases

#### 3.3.1. In High-Income Countries 

In the USA, the consumption of palm oil was not significantly correlated with the mortality of the burden of CBVDs or DM. However, the consumption of soya oil in the USA showed a significant positive correlation (*p* ≤ 0.033) with the mortality burden of DM, while there was no significant correlation with that of CBVDs. In Japan, the consumption of palm oil showed a significant negative correlation with the mortality burden of CBVDs and DM (*p* ≤ 0.042 and *p* ≤ 0.005, respectively), while the consumption of soya oil was not significantly correlated with the mortality burdens of CBVDs and DM. In Australia and Switzerland, the consumption of soya oil showed a significant (*p* ≤ 0.014 and *p* ≤ 0.023, respectively) positive correlation with the mortality burden of CBVDs but not with DM, while the consumption of palm oil was not significantly correlated with the mortality burdens of CBVDs and DM (Figure 4).

#### 3.3.2. In High Middle-Income Countries 

In Turkey and Saudi Arabia, the consumption of palm and soya oils showed a nonsignificant correlation with the mortality burdens of CBVDs and DM except for Saudi Arabia, where the consumption of soya oil showed a significant (*p* ≤ 0.031) negative correlation with the mortality burden of DM. In Malaysia, the consumption of palm and soya oils showed a nonsignificant correlation with the mortality burdens of CBVDs and DM (Figure 5).

#### 3.3.3. In Middle-Income Countries

In Brazil, the consumption of palm oil showed a nonsignificant correlation with the mortality burdens of CBVDs and DM, while the consumption of soya oil showed a significant (*p* ≤ 0.030) negative correlation with the mortality burden of CBVDs, and nonsignificant correlation with that of DM. In China, neither the mortality burden of CBVD nor DM showed a significant correlation with the consumption of either palm or soya oil. In Jordan, the consumption of palm oil showed a significant (*p* ≤ 0.029) positive correlation with the mortality burden of DM but a nonsignificant correlation with that of CBVDs, while the consumption of palm oil showed a nonsignificant correlation with the mortality burdens of CBVDs and DM. In Indonesia, the consumption of soya oil showed a significant (*p* ≤ 0.028) positive correlation with the mortality burden of CBVDs, but not with that of DM, while the consumption of palm oil showed a nonsignificant correlation with the mortality burdens of CBVDs and DM (Figure 6).

## 4. Discussion

Previous reports have suggested that DM is a risk factor for the incidence of CBVDs [5,28], and that fat intake is involved in the incidence and prevalence of CBVDs and DM [13,19]. However, there are no previous reports about the relationship between the consumption of palm or soya oils with the mortality burden of either CBVDs or DM.

In high-income countries, it is obvious that the mortality burden of CBVDs in the USA remained unstable, which could be due to the increase in the mortality burden of CBVDs among adults in the USA [6]. Conversely, the mortality burden of CBVDs declined in Japan, Australia, and Switzerland. However, there were no significant positive correlations between the mortality burdens of CBVDs with time in all high-income countries. Perhaps high-income countries were successful in lowering the mortality burden of CBVDs due to the availability and accessibility of primary healthcare [8,29]. Similarly, the trend of the mortality burden of DM in all high-income countries continued to decline from one year to another. In addition, there was either a nonsignificant correlation or significant negative correlation between time and the mortality burden of DM in all high-income countries, which could indicate good control measures due to the availability and accessibility of primary health care [8,29].

The consumption of palm oil in all high-income countries increased, while the consumption of soya oil in the USA was the highest, which is normal since the USA is the second biggest producer of soya oil in the world [23,30]. However, the consumption of soya oil in the USA, Australia, and Switzerland declined with time except for Japan, where the consumption of soya oil tended to increase. Nonetheless, the consumption of palm oil in Japan was positively correlated with time, which indicates an increasing trend in the consumption of palm oil. In addition, the consumption of soya oil was not significantly correlated, either positively or negatively, with time. Accordingly, the former findings could indicate that the consumption of palm oil is more popular and traditionally consumed than that of soya oil in high-income countries.

The findings indicated that the consumption of soya oil was positively correlated to the mortality burden of CBVDs in Australia and Switzerland, while palm oil showed either a nonsignificant or significant negative correlation with the mortality burden of CBVDs in all high-income countries. Regarding DM, the consumption of soya oil was significantly positively correlated with the mortality burden of DM in the USA, while the consumption of palm was either nonsignificantly correlated or significantly negatively correlated with the mortality burden of DM. Although the former findings indicated that the consumption of soya oil could be a possible risk factor to the mortality burdens of CBVDs and DM in some high-income countries, there was no experimental evidence to support such a conclusion. Perhaps investigating the role of the pathophysiology of age-related CBVDs [31] or DM could lead to a better understanding of the mortality burdens of those diseases, particularly that aging is one of the reasons that contribute to reducing life expectancy among the elderly in Japan [32]. Although no interpretations could be suggested in this regard, performing well-controlled and longer clinical trials could disclose the effects or adverse effects of consuming either palm or soya oil. Further epidemiological studies should evaluate the trend of mortality burdens of CBVDs and DM over longer periods, which is one of the limitations of our study due to the unavailability of data for certain years within either the period of 2005 to 2010 or 2010 to 2016. In addition, other covariates such as smoking, fatty foodstuff, and age should be considered in the future during the evaluation of the association between the mortality burdens of CBVDs and DM and the food consumption of edible oils.

In high middle-income countries, the trend of mortality burdens of CBVDs and DM declined in all high middle-income countries except that the mortality burden of DM increased in Turkey, which could be due to DM being more common among Turkish people according to the international standards [33,34,35]. On the other hand, there was an upward trend in palm oil consumption in Turkey, Saudi Arabia, and Malaysia. Meanwhile, the consumption of soya oil in Saudi Arabia and Malaysia followed an upward trend, but it followed a downward trend in Turkey, which could be due to Turkey occupying the third global rank of olive oil production [23]. In Malaysia, the increase in the consumption of palm oil is usual in terms of the consumption and production of palm oil since Malaysia is the fourth biggest consumer and second biggest producer of palm oil, while the increase in the consumption of palm oil in Saudi Arabia could be due to its acceptability by consumers. The perception and awareness about the consumption of palm oil in Saudi Arabia have not yet been reported. However, the comparison in terms of the consumed quantities did not change the reality of the huge consumption of palm oil in these countries [23]. Although Turkey occupies the rank of the third biggest global producer of olive oil [23], the consumption of palm oil increased, which could be due to the lower price of palm oil compared to the other vegetable oils [36]. The former findings could indicate that the consumption of palm oil has been superior to that of soya oil in all high middle-income countries. The findings of the correlations between the mortality burden of either CBVDs or DM with the consumption of palm and soy oils were negative or nonsignificant, indicating that the primary health care in high middle-income countries was relatively good to reduce the mortality burdens of CBVDs and DM [37,38,39,40,41]. However, the primary health in these countries has been more focused on infant and child health [42]. Perhaps further studies should focus on investigating the relationship between the mortality burdens of CBVDs and DM and other risk factors such as age, smoking, and fat consumption. 

In middle-income countries, the mortality burdens of CBVDs and DM in Brazil, China, and Jordan declined, in contrast to Indonesia, where these burdens increased. However, the correlations between the mortality burdens of CBVDs and DM with time in middle-income countries were either nonsignificant or significantly negative, indicating that the mortality burden of either DM or CBVDs in the middle-income countries could be controlled toward steady-state, despite that Indonesia has limited resources to address the increase of noncommunicable epidemics [43]. On the other hand, the trends of consumption of palm and soya oils in the middle-income countries were unstable. However, the consumption of palm oil in all these countries has been superior to soya oil, which could be due to the cheaper price of palm oil [36]. In Jordan, the consumption of palm oil showed a positive correlation with the mortality burden of DM, while the consumption of soya oil showed a significant positive correlation to the mortality burden of CBVDs in Indonesia, which could be related to the differences in the availability and accessibility of primary health prevention between one country to another.

It could be noted from the consumption of oils in the high-, high–middle and middle-income countries, palm oil consumption was dominant, which could be due to palm oil’s advantages over other oils, such as its resistance to oxidation, stability at frying temperatures, the longer shelf life of the finished product, the ability to be blended with other edible oils [30], and cheaper prices. In addition, palm oil is free of the hazards associated with trans-fatty acids, which has reduced the use of hydrogenated soybean oil [30]. Moreover, the popularity of palm oil among consumers could come from the good perception and awareness about their healthy and safe consumption, despite the national campaigns undertaken to compulsorily remove palm oil from manufactured products, to be replaced by hydrogenated vegetable oils that increase the risk of trans-fatty acids [44]. 

The underlying mechanisms of associations between soya and palm oils with the mortality of CVBDs and DM are not clear, and no evidence could be extracted from the literature about the underlying mechanisms. Therefore, conducting a longer epidemiological or longitudinal study on palm oil consumption is recommended.

Since the diet is a potentially modifiable factor influencing CBVDs and DM, this study tried to answer relevant questions regarding correlations of food-use consumption of soya and palm oils with mortality burdens of CBVDs and DM. This study relies on country-level data on the consumption of soya and palm oil and mortality burdens of diseases, of which the quality may vary between the countries assessed. However, the analyses do not address any potential sources of confounding, which may be related to differences in population demographics, differences in disease detection or treatment, or globalization. Accordingly, perhaps the disagreement between increasing consumption and mortality burden of disease between the countries could be due to uncontrolled confounding related to differences in health characteristics between the countries. Moreover, the sample size in the current study was too small, and the data from different years in each group led to heterogeneity between the groups. Accordingly, future studies should consider such limitations to elaborate on this work.

## 5. Conclusions

Among the high-income countries, a positive relationship was indicated between the consumption of soya oil and the mortality burden of CBVDs in Australia and Switzerland, as well as the mortality burden of DM in the USA. In high middle-income countries, there was no correlation between the mortality burdens of CBVDs and DM with the consumption of either palm or soya oil. In middle-income countries, the consumption of palm oil had a positive relationship with the mortality burdens of DM in Jordan, while the consumption of soya oil had a positive relationship with the mortality burden of CBVDs in Indonesia. Nonetheless, perhaps the differences in specifications of the offered primary health care and/or prevention, income and the age-related risk factors could play a role in the mortality burden of either CBVD or DM in the different countries, which should be considered in making a firm decision about the possible contribution of the consumption of either palm or soya oil worldwide. Although it is revealed that the consumption of palm oil in all countries is still dominant or even superior to that of soya oil, however, the consumption of oils in each country seems to be controlled by either the production of oils or it might be the perception and acceptability of consumers to certain oil. Accordingly, conducting cohort studies to monitor the consequence of exposure to either palm or soya oil over a long period as well as to conduct controlled crossover clinical trials to elucidate the effects of consuming palm and soya oils is strongly recommended.

## Figures and Tables

**Figure 1 ijerph-17-05410-f001:**
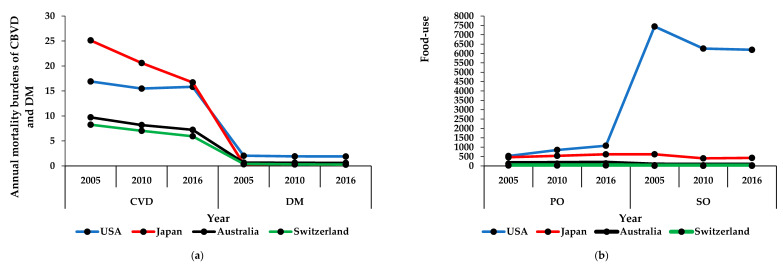
Mortality burdens of diseases and consumption of oils in high-income countries. (**a**) The annual mortality burdens of cerebrovascular disease and diabetes mellitus (number of deaths per 100,000 population per year); (**b**) the annual food-use consumption of palm and soya oils (tons per capita per year). The results are expressed in the original values in 2005, 2010, and 2016 that were obtained from the specified data sources. USA, United States of America; CBVD, cerebrovascular disease; DM, diabetes mellitus; PO, palm oil; SO, soya oil.

**Figure 2 ijerph-17-05410-f002:**
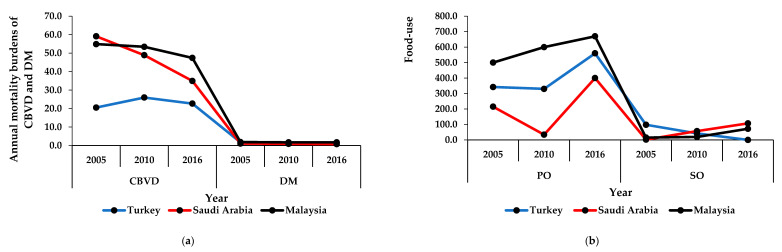
Mortality burdens of diseases and consumption of oils in high middle-income countries. (**a**) The annual mortality burdens of cerebrovascular disease and diabetes mellitus (number of deaths per 100,000 population per year); (**b**) the annual food-use consumption of palm and soya oils (tons per capita per year). The results are expressed in the original values in 2005, 2010, and 2016 that were obtained from the specified data sources. CBVD: cerebrovascular disease, DM: diabetes mellitus, PO: palm oil, SO: soya oil.

**Figure 3 ijerph-17-05410-f003:**
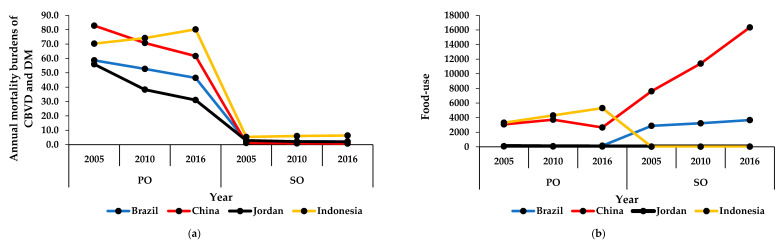
Mortality burdens of cerebrovascular diseases and consumption of oils in middle-income countries. (**a**) The annual mortality burdens of cerebrovascular disease and diabetes mellitus (number of deaths per 100,000 population per year); (**b**) the annual food-use consumption of palm and soya oils (tons per capita per year). The results are expressed in the original values in 2005, 2010, and 2016 that were obtained from the specified data sources. CBVD: cerebrovascular disease, DM: diabetes mellitus, PO: palm oil, SO: soya oil.

**Figure 4 ijerph-17-05410-f004:**
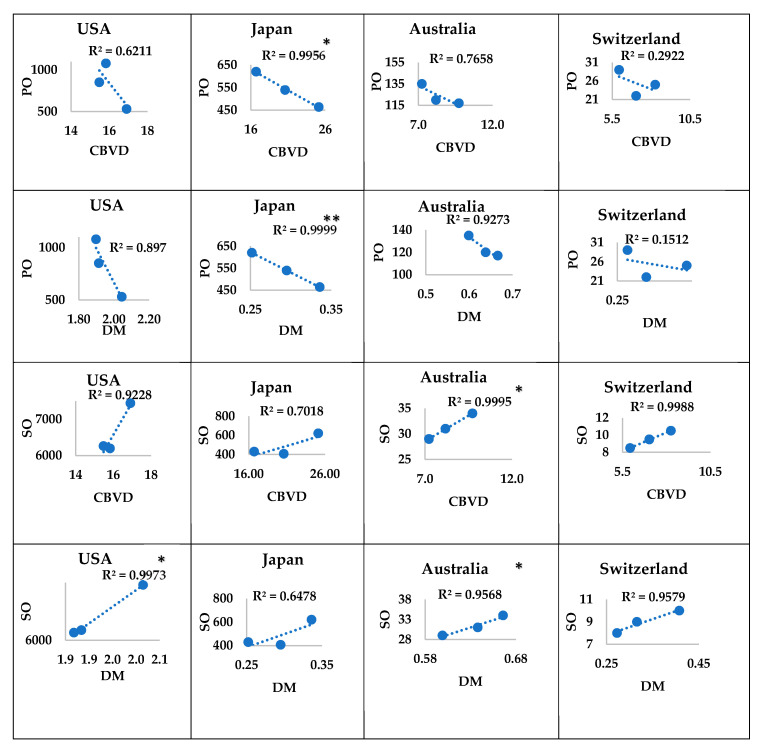
Correlations of the mortality burdens with consumption of oils in high-income countries. Orignal values of food-use consumption of palm or soya oil (*y*-axis) were plotted against those of mortality burdens of CBVDs or DM (*x*-axis). * *p* ≤ 0.05 and ** *p* ≤ 0.01 denote significant correlations. Upward trendline denotes a positive correlation, while downward trendline denotes a negative correlation. R^2^ denotes the correlation coefficient. The sample size was three readings of mortality burdens or consumption of oils in 2005, 2010, and 2016. USA: United States of America, CBVD: cerebrovascular disease, DM: diabetes mellitus, PO: palm oil, SO: soya oil.

**Figure 5 ijerph-17-05410-f005:**
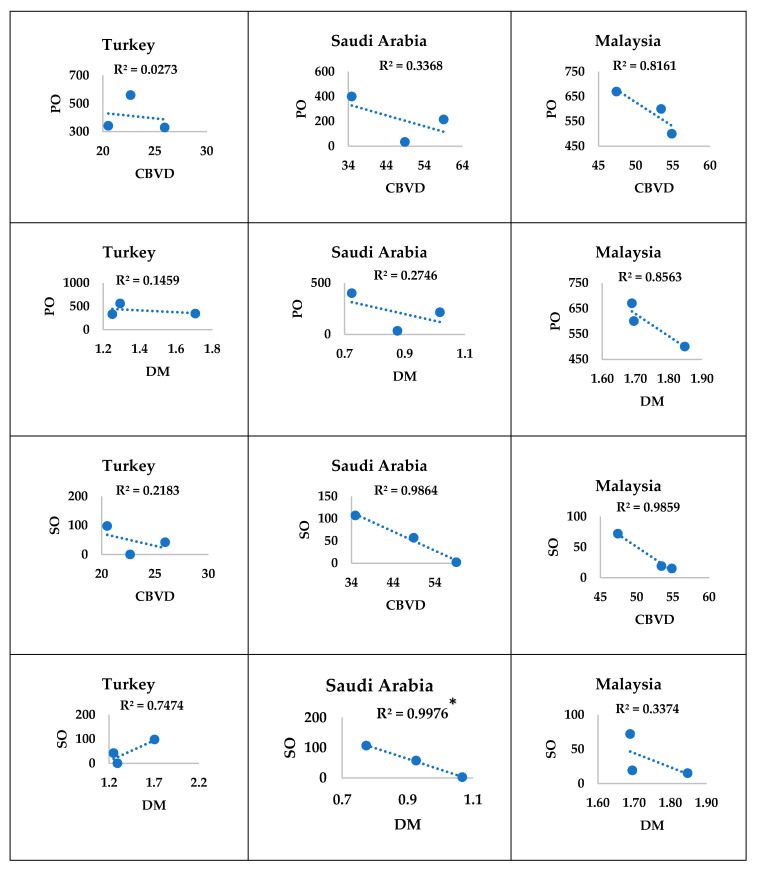
Correlations of the mortality burdens with consumption of oils in high middle-income countries. Orignal values of food-use consumption of palm or soya oil (*y*-axis) were plotted against those of the mortality burden of CBVDs or DM (*x*-axis). * *p* ≤ 0.05 denotes significant correlations. Upward trendline denotes a positive correlation, while downward trendline denotes a negative correlation. R^2^ denotes the correlation coefficient. The sample size was three readings of mortality burdens or consumption of oils in 2005, 2010, and 2016. CBVD: cerebrovascular disease, DM: diabetes mellitus, PO: palm oil, SO: soya oil.

**Figure 6 ijerph-17-05410-f006:**
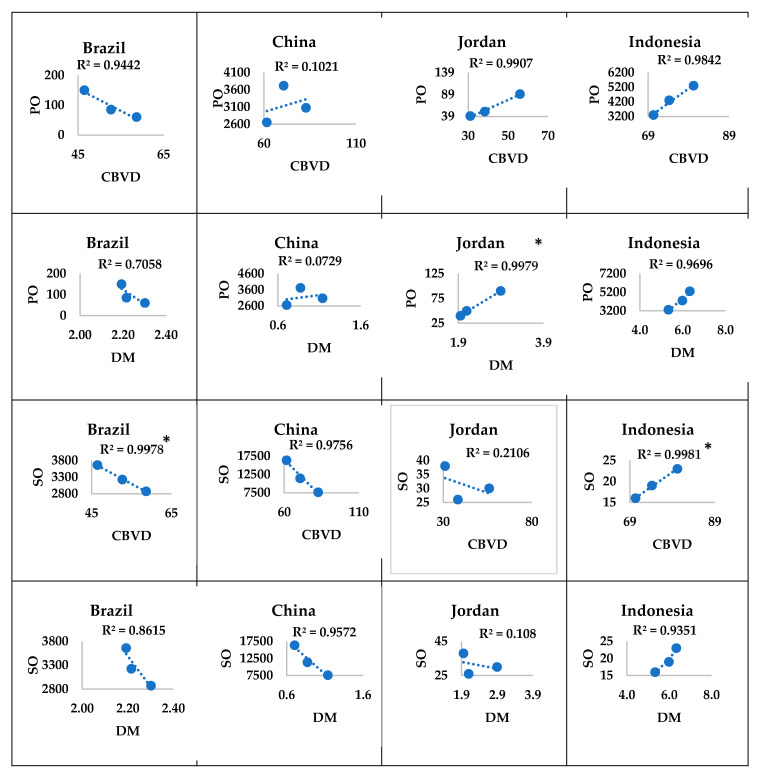
Correlations of the mortality burdens with consumption of oils in middle-income countries. Orignal values of food-use consumption of palm or soya oil (*y*-axis) were plotted against those of the mortality burden of CBVDs or DM (*x*-axis). * *p* ≤ 0.05 denotes significant correlations. Upward trendline denotes a positive correlation, while downward trendline denotes a negative correlation. R^2^ denotes the correlation coefficient. The sample size was three readings of mortality burdens or consumption of oils in 2005, 2010, and 2016. CBVD: cerebrovascular disease, DM: diabetes mellitus, PO: palm oil, SO: soya oil.

**Table 1 ijerph-17-05410-t001:** Original values of the mortality burden of cerebrovascular diseases and food-use consumption of oils in high-income countries.

	USA			Japan			Australia			Switzerland		
	2005	2010	2016	2005	2010	2016	2005	2010	2016	2005	2010	2016
CBVD	16.91	15.48	15.83	25.13	20.60	16.71	9.73	8.18	7.23	8.26	7.03	5.94
DM	2.05	1.91	1.90	0.34	0.30	0.25	0.67	0.64	0.60	0.41	0.32	0.27
PO	531.00	852.00	1080.00	464.00	539.00	620.00	117.00	120.00	135.00	25.00	22.00	29.00
SO	7441.00	6264.00	6198.00	621.00	407.00	430.00	34.00	31.00	29.00	10.00	9.00	8.00

CBVD: cerebrovascular disease, DM: diabetes mellitus, PO: palm oil, SO: soya oil, USA: United States of America

**Table 2 ijerph-17-05410-t002:** Original values of the mortality burdens of diseases and food-use consumption of oils in high middle-income countries.

	Turkey			Saudi Arabia			Malaysia		
	2005	2010	2016	2005	2010	2016	2005	2010	2016
CBVD	20.52	25.96	22.67	59.11	48.89	34.94	54.86	53.43	47.42
DM	1.71	1.25	1.29	1.02	0.88	0.72	1.85	1.70	1.69
PO	342.00	330.00	560.00	215.00	34.00	400.00	500.00	600.00	670.00
SO	98.00	42.00	0.00	2.00	57.00	107.00	15.00	19.00	72.00

CBVD: cerebrovascular disease, DM: diabetes mellitus, PO: palm oil, SO: soya oil.

**Table 3 ijerph-17-05410-t003:** Original values of the mortality burden of diseases and food-use consumption of oils in middle-income countries.

	Brazil			China			Jordan			Indonesia		
	2005	2010	2016	2005	2010	2016	2005	2010	2016	2005	2010	2016
CBVD	58.67	52.70	46.47	82.82	70.75	61.64	55.95	38.28	31.03	70.33	74.18	80.21
DM	2.30	2.22	2.19	1.14	0.87	0.71	2.89	2.10	1.95	5.33	5.98	6.33
PO	60.00	85.00	150.00	3074.00	3717.00	2650.00	90.00	50.00	40.00	3300.00	4300.00	5300.00
SO	2871.00	3225.00	3660.00	7607.00	11400.00	16350.00	30.00	26.00	38.00	16.00	19.00	23.00

CBVD: cerebrovascular disease, DM: diabetes mellitus, PO: palm oil, SO: soya oil.

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
