# Peer review of "Correlation of Mortality Burdens of Cerebrovascular Disease and Diabetes Mellitus with Domestic Consumption of Soya and Palm Oils"

_ijerph, 2020, doi:10.3390/ijerph17155410_

Round 1

Reviewer 1 Report

line 34. There is too much space between "and" and "food"

line 40. "burden should read "burdens"

Other places where the singular "burden" should be replaced by the plural "burdens": line 298, line 320, line 472 and line 476.

Places where the plural "burdens" should be replaced by the singular "burden": line 364, line 366, line 394, line 395 and line 401.

line 480. The beginning of the line should read "of the mortality burden of either DM ..."

Other changes required: "high income" lacks a hyphen on line 177 and in Table 1, 2nd column.  Also, "middle income" lacks a hyphen on line 176, line 177, line 186 and Table 1, 4th column.

line 514. Replace "evidence" by "piece of evidence".

Author Response

Thank you for the reviewer. Please, see the attachment

Reviewer 2 Report

The purpose of this manuscript is to summarize the correlations of cerebrovascular diseases (CBVD) and diabetes mellitus (DM) with consumption of soya and palm oils in 2005, 2010 and 2016 among 11 countries. In general, it is an interesting topic since diet is a potentially modifiable factor influencing CBVD and DM. Despite the limitation in the review structure, the summary performed still can answer relevant questions regarding correlations of soya and palm oils with prevalence of CBVD and DM. However, I believe that summaries can be performed in different ways to make the manuscript clear and straightforward. Also, some clarifications in the methods and a full disclosure of the study limitations are required.

Introduction

  • Line 63-65: Currently increasing evidence shows there are associations between DM and Alzheimer’s disease; the information with AD is not accurate. In addition, it seems that these several sentences regarding neurodegeneration are not related to the topic of this paper.
  • Line 102: since this paper did not summarize any information within the timeframe, it should say specific time point.

Methods

  • Line 109-112: it is not clear to say “annual per capita deaths per 100,000 population”, same with description of food consumption.
  • Line 123: how the sampled countries were randomly selected should be described including the specific random method.
  • Line 131: what is “risk-standardized death rate” and what kind of risk should be clear.
  • Line 132-137: the formula of converting from number of per capital to per 100,000 population and shifting of units should be provided. In addition, the oil consumption should not be converted into per 100,000 population; amount per capita is more common and meaningful to display.
  • Line 55-58: since the sample size is too small and each data from different year in each group, there must be large heterogeneity between groups, and it is not appropriate to conduct ANOVA. Also, the power of these tests should be taken into attention.
  • Line 59: why standard error was presented rather than standard deviation?
  •  

Results

  • Figure 1-2: since there is only three data in each country and the number of observations is even less than the number of comparison groups (e.g. 4 high-income countries), it is problematic to conduct ANOVA and the results won’t be robust.
  • Table 1-2: it is meaningless to average the death rate and oil consumption among different countries and different years.
  • Could not understand why authors using statistical methods to compare death rate and oil intake by combining data from different countries and years. I think the aim of this review is to explore the correlation between mortality of oil consumption and CBVD as well as DM. However, authors showed too many comparison results and the comparisons are statistically problematic. Recommend to just describe the actual number and trends of mortality and oil consumption simply like Figure 3-5, and cut Figure 1-2 and table 1-2 off.
  • Figure 3-5: it is misleading to use bar charts to show one-point data. Recommend to use line charts, where year is on the x-axis and mortality or consumption is on the y-axis, grouped by different countries in different colours.
  • Table 3-5: if line charts were used in the last comment, it is no need to show table 3-5 anymore.
  • Figure 6-8: it is not right and useless to correlate mortality rate or food consumption with time. On the contrary, people more concern the trend of them in each country which is already shown in figure 3-5.
  • Result 3.5: correlation results could be shown in figures a way like figure 8 where the x-axis should be food consumption and y-axis should be mortality.
  • Line 365: Specific p values should be provided throughout the whole paper.

Discussion and Conclusion

  • Discussion and conclusion should be updated accordingly. The underlying mechanisms of associations between soya/palm oils and mortality of CVBD and DM will be more interested to readers. Limitations should be disclosed.

Abstract

  • Line 35: specific years should be indicated rather than “within 2005 and 2016”.

Generally, the written English is poor. Grammar and punctuations should be double checked.

Author Response

(The authors gave the same response as above.)

Round 2

Reviewer 2 Report

The authors have made substantial revisions, which significantly improved the quality of the manuscript. However, I still have a few concerns as described below.

  1. With regard to the term “number of death cases per capita per 100,000 population” in Line 107, it is ambiguous to express like this. Either per capita or per 100,000 population should be in the sentence rather than both. Please refer to the definition from WHO (https://www.who.int/data/gho/indicator-metadata-registry/imr-details/3130).
  2. Considering that the annual food consumption per capital is already standardized by general population in each country, please provide justification of “product was divided by the total population in millions of each country per a specific year” in formula (3). In addition, express like “total per capita population” is not accurate.
  3. As mentioned before, if authors insist to convert the oil consumption to per 100,000 population for matching with mortality rate, please provide sufficient evidence or publications like this. Annual food consumption per capital is recommended to be used in comparisons between countries by WHO (https://www.who.int/nutrition/topics/3_foodconsumption/en/) and can be correlated with mortality rate.
  4. Please double check your grammar and spell. Typos are everywhere, e.g. anuual in formula 3 (Line 148).
  5. In figure 1-3, lines connecting different countries are meaningless which should be shifted to connecting different years to show trends. Please re-format them referring to an example from WHO (https://www.who.int/nutrition/topics/3_foodconsumption/en/index6.html).
  6. In figure 4-6, the meaning of R square should be indicated in figure captions. In addition, p values for each correlation should be indicated via either showing specific p values in corresponding panels or using star systems (e.g. * denotes p<0.05).
  7. In abstract, please clarify why only positive correlations were highlighted while many negative correlations were not mentioned.

Author Response

This manuscript is a resubmission of an earlier submission. The following is a list of the peer review reports and author responses from that submission.

Round 1

Reviewer 1 Report

line 42 “to the mortality burdens” is better than “in the mortality burdens”

line 79 Delete the comma after “whereas”

line 233 Should read “from one year to another”

line 235 Insert “significant” after “statistically”

line 272 Insert full stop (a period) at aend of line

line 354 soya.oil The full stop (period) should be moved to appear after “oil”, leaving a space between “soya” and “oil”

line 440 “Malaysia had a higher mortality burden” (not “burdens”), and also “and a higher” at end of sentence

line 442 “except that” instead of “except for”

line 458 Replace “due to that Malaysia producing soya oil” by “due to the fact that Malaysia is a producer of soya oil”

line 469 “Soya oil” should read “soya oil”

line 474 Should read “burden of CBVD”, not “burdens of CBVD”

line 474 Delete “continued”

line 482 Delete “still”

line 486 Delete “the” before “primary”

line 489 Insert a comma after “dominant”

line 492 Replace “free of the hazards of containing trans-fatty acids” by “free of the hazards associated with trans-fatty acids”

line 517 Delete “at all”

Additional comment. Usually “middle-income” is written with a hyphen, but there are several instances where “middle income” is written without a hyphen. In addition, “high-income” is usually written with a hyphen, but there are several instances where “high income” is written without a hyphen. Consistency is important.

A further comment. The term "metric tons" in the original manuscript has been changed to "tons". Is that correct? A "ton" is an English (Imperial) measure (=2000 pounds). If "metric tons" is really meant (each metric ton=1000 kilograms), then it should be left at that, or replaced with "tonnes", which is another way of writing "metric tons".

Reviewer 2 Report

The authors did not sufficiently respond to the comments from the first round of peer review. Changes in the document do not seem to address the comments, and it appears there were no changes to the analyses performed. Therefore, the issues from the first submission remain unresolved and the article is not publishable in it's current state.